# Positive Associations of Dietary Intake and Plasma Concentrations of Vitamin E with Skeletal Muscle Mass, Heel Bone Ultrasound Attenuation and Fracture Risk in the EPIC-Norfolk Cohort

**DOI:** 10.3390/antiox10020159

**Published:** 2021-01-22

**Authors:** Angela A. Mulligan, Richard P. G. Hayhoe, Robert N. Luben, Ailsa A. Welch

**Affiliations:** 1Department of Public Health and Primary Care, Institute of Public Health, University of Cambridge, Cambridgeshire CB1 8RN, UK; robert.luben@phpc.cam.ac.uk; 2NIHR BRC Diet, Anthropometry and Physical Activity Group, MRC Epidemiology Unit, University of Cambridge, Cambridgeshire CB2 0AH, UK; 3Department of Epidemiology and Public Health, Norwich Medical School, Faculty of Medicine and Health Sciences, University of East Anglia, Norwich NR4 7TJ, UK; R.Hayhoe@uea.ac.uk (R.P.G.H.); A.Welch@uea.ac.uk (A.A.W.)

**Keywords:** sarcopenia, frailty, skeletal muscle, bone density status, fracture risk, vitamin E

## Abstract

The prevalence of sarcopenia, frailty and fractures is increasing. Prevention options are limited, but dietary factors including vitamin E have the potential to confer some protection. This study investigated cross-sectional associations between dietary and plasma concentrations of vitamin E with indices of skeletal muscle mass (SMM) (*n* = 14,179 and 4283, respectively) and bone density (*n* = 14,694 and 4457, respectively) and longitudinal fracture risk (*n* = 25,223 and 7291, respectively) in European Prospective Investigation Into Cancer and Nutrition (EPIC)-Norfolk participants, aged 39–79 years at baseline. Participants completed a health and lifestyle questionnaire, a 7-day diet diary (7dDD) and had anthropometric measurements taken. Fat-free mass (as a SMM proxy) was measured using bioimpedance and bone density was measured using calcaneal broadband ultrasound attenuation (BUA) and incident fractures over 18.5 years of follow-up. Associations between indices of SMM, BUA and fracture risk were investigated by quintiles of dietary vitamin E intake or plasma concentrations. Positive trends in SMM indices and BUA were apparent across dietary quintiles for both sexes, with interquintile differences of 0.88–1.91% (*p* < 0.001), and protective trends for total and hip fracture risk. Circulating plasma α- and γ-tocopherol results matched the overall dietary findings. Dietary vitamin E may be important for musculoskeletal health but further investigation is required to fully understand the relationships of plasma tocopherols.

## 1. Introduction

The prevalence of sarcopenia (low levels of muscle strength, muscle quantity/quality and physical function [1]), frailty and fractures is increasing in our aging society. In the UK, the number of older people is growing; in 2018, there were 1.6 million people aged 85 years and over; by mid 2043, this is projected to nearly double to 3 million [2]. In parallel, the number of sarcopenic patients will dramatically increase, adding to the already considerable resultant public health issues [3]. A recent prospective study by Sousa et al. [4] found that sarcopenia was independently associated with hospitalisation costs and with an estimated increase of 34% for patients aged ≥65 years. Approximately 520,000 fragility fractures occurred in the UK in 2017, with fracture-related costs of GBP 4.5 billion; these numbers are estimated to increase by 26.2% and 30.2%, respectively, by 2030 [5]. Losses in bone density and skeletal muscle mass and strength occur gradually from the age 30 years, with increasing rates of loss in those over the age of 60 years [6,7]. These conditions are currently difficult to treat and, therefore, maintaining skeletal muscle and bone health during aging is important.

It has long been established that a close physiological relationship exists between muscle and bone, which changes with aging, but more recently it has become apparent that this is not solely related to mechanical function [8,9]. Factors, such as myokines, that are secreted by muscle, including insulin-like growth factor 1 and fibroblast growth factor 2, have paracrine and endocrine effects which can affect bone repair and metabolism [10,11]. Additionally, bone secretes factors such as osteocalcin and connexin 43, that have direct effects on muscle [12,13]. Studies have shown that both sarcopenia and frailty are risk factors for fractures and falls [14,15,16,17,18].

Known determinants of muscle and bone aging [19] include modifiable lifestyle risk factors, such as cigarette smoking, low physical activity and poor diet [20]. Limited research exists, mainly in older adults, which suggests that vitamin E, a lipid-soluble, anti-oxidant vitamin, may also be protective with respect to muscle mass and frailty [21,22,23,24,25,26], as skeletal muscle is the organ with the highest consumption of oxygen in the body. Positive associations of vitamin E intake with bone mineral density (BMD) and fracture risk have also been reported in both men and women [27,28,29,30]. The major forms of vitamin E in food are the α- and γ-tocopherols, and thus these are found in greater abundance than other tocopherols and tocotrienols in tissues. The predominant form of vitamin E in the body is α-tocopherol, which has tended to be the focus of research, although some research has been carried out on other tocopherols, in particular, γ-tocopherol [31,32]. A number of mechanisms have been suggested as to how vitamin E may slow down aging of skeletal muscle [33,34,35]. Reduction in oxidative stress is thought to be a mechanism by which vitamin E homologues protect bone but it has been reported that α- and γ-tocopherol have opposing inflammatory functions and may uncouple bone turnover, such as by increasing bone resorption without affecting bone formation [36,37].

The current study therefore aimed to investigate the potential associations of reported dietary vitamin E intake (α-tocopherol equivalents), as well as plasma concentrations of both α- and γ-tocopherol and the ratio of α:γ-tocopherol, with measures of skeletal muscle mass (SMM) and bone density status concurrently, in a large cohort general population cohort of middle-aged and elderly men and women. Additionally, both dietary and plasma concentrations of vitamin E were examined in relation to fracture risk during 18.5 years of follow-up. 

## 2. Materials and Methods

### 2.1. EPIC-Norfolk Study Design

The Norfolk cohort of the European Prospective Investigation Into Cancer and Nutrition (EPIC-Norfolk) is part of the Europe-wide EPIC study, which involves over half a million people in ten countries [38] and was initially designed to investigate diet and the risk of developing cancer. Details of cohort recruitment, data collection and participant characteristics have been published previously [39]. In brief, participants aged between 39 and 79 years were recruited from General Practitioners’ surgeries, based in the rural areas of Norfolk and market towns as well as the city of Norwich, from 1993 to 1997. Since virtually all the population of the UK are registered with a general practice through the National Health Service, general practice age sex registers act as a population sampling frame. This cohort at baseline was comparable to the UK national population with regard to many characteristics, including age, sex and anthropometry measurements, but it had a lower proportion of current smokers [40]. The study was approved by the Norfolk District Health Authority Ethics Committee (98CN01) and all participants gave written informed consent, according to the Declaration of Helsinki. Of the 30,445 men and women who consented to participate in the study (39% response rate), 25,639 attended a baseline health examination (1HE) between 1993 and 1997. Of these, 15,028 attended a second health examination (2HE) between 1998 and 2000. 

### 2.2. Measurements of Body Composition

#### 2.2.1. Height, Weight and BMI

At both health examinations, a trained nurse measured weight (to the nearest 0.1 kg) and height (to the nearest 0.1 cm), with participants wearing light clothing and no shoes. Body mass index (BMI) was calculated from these measurements as weight in kilograms divided by height squared in metres (kg/m^2^). 

#### 2.2.2. Indices of Fat-Free Mass (FFM)

Bioelectrical impedance analysis (BIA) was carried out at 2HE, using a standardised protocol (Bodystat, Isle of Man, UK), suitable for use in large field-based studies and shown to be a valid [41] and reliable [42] measure of body composition. The TANITA Body Fat Monitor/Scale TBF-531 BIA analyser (Tanita UK Ltd., Middlesex, UK) calculated body density (BD) from total weight (Wt) in kilograms, height (Ht) in centimeters, and impedance (Z) in ohms, using the following standard regression formulae: BD in men = 1.100455 − 0.109766 × Wt × Z ÷ Ht^2^ + 0.000174 × Z; BD in women = 1.090343 − 0.108941 × Wt × Z ÷ Ht2 + 0.00013 × Z. Fat-free mass (FFM) in kilograms was then calculated: FFM = Wt − ((4.57 ÷ BD − 4.142) × Wt), which is an estimate of the total mass of nonfat compartments of the body—i.e., metabolic tissue, water, and bone. In an effort to scale for differences in skeletal muscle mass with increasing body weight, scaled for height, FFM standardised by BMI (FFM_BMI_) was calculated as FFM divided by BMI, according to the method suggested by Studenski [43].

#### 2.2.3. Bone Density Assessment 

Quantitative ultrasound measurements of the calcaneus (heel bone) were taken at 2HE, using a contact ultrasound bone analyser (CUBA) device (McCue Ultrasonics) following standard protocols. Broadband ultrasound attenuation (BUA) (dB/MHz) measurements were taken at least in duplicate for each foot of the participant, and the mean of the left and right foot measures was used for analysis. Each of the five CUBA devices used in the study was calibrated daily with its physical phantom. In addition, calibration between devices was checked monthly using a roving phantom (CV 3.5%). The CUBA method of bone density assessment has been shown to be capable of predicting fracture risk [44,45], and is cheaper and simpler to conduct in general practice settings compared with the gold-standard of dual-energy X-ray absorptiometry (DEXA).

#### 2.2.4. Fracture Incidence

Self-reported fracture was recorded using questionnaires at both 1HE and 2HE, and incident fracture was ascertained using linkage to the East Norfolk Health Authority database (ENCORE) of hospital attendances by Norfolk residents [46]. Incidence of all osteoporotic fractures in the cohort, up to the end of March 2018, was determined by retrieving data using each participant’s National Health Service (NHS) number and searching for events logged using the International Classification of Diseases (ICD) 9 and 10 diagnostic codes for osteoporotic hip, spine or wrist fractures. 

### 2.3. Measurement of Vitamin E Intake

Dietary intakes at 1HE were assessed using 7-day diet diaries (7dDDs), which were completed by 25,507 participants, detailing all food and drink consumed, together with the portion sizes. Data Into Nutrients for Epidemiological Research (DINER) software was used to enter the dietary information provided by the 7dDDs [47], which was then checked and processed by nutritionists to obtain nutrient data using DINERMO [48]. Vitamin and mineral supplements recorded in the 7dDD were quantified using the Vitamin and Mineral Supplement (ViMiS) database [49].

### 2.4. Blood Analysis

At 1HE, a 42 mL sample of blood was collected in citrated and plain monovettes and stored in a refrigerator. The next day, blood samples were processed and stored at −196 °C as plasma and serum. Serum cholesterol was determined for the full cohort in a Norfolk laboratory using a RA 1000 Diagnostics (Bayer, Basingstoke, UK) instrument, and cohort concentrations ranged from 2.10 to 12.40 mmol/L. The vitamins α- and γ-tocopherol were analysed on a cohort subset that consisted of a series of previous case-control studies, where cases were defined by incident cardiovascular disease or cancer and four matched, disease-free controls. Plasma concentrations were analysed at IARC, Lyon (France), using high-performance liquid chromatography for the vitamins. In our analyses, concentrations for α-tocopherol ranged from 0.71 the 106.54 μmol/L and from 0.03 to 9.85 μmol/L for γ-tocopherol; we excluded one participant for whom the ratio of α-tocopherol to γ-tocopherol was greater than 1000). Plasma α- and γ-tocopherol concentrations were adjusted for cholesterol, as this is seen as a more reliable marker for vitamin E nutritional status [50] since tocopherols are transported via circulation through lipoproteins; adjusted concentrations are presented in μmol/mmol, calculated by dividing the plasma tocopherol concentrations (μmol/L) by total cholesterol (mmol/L). 

### 2.5. Measurement of Confounding Variables

Data collected via two self-administered health and lifestyle questionnaires (HLQ1 and HLQ2), before the 1HE and 2HE, respectively, were used to establish classification of a number of variables. Family history of osteoporosis was categorised as yes or no; menopausal status (women only (2HE)) was categorised as premenopausal, perimenopausal (<1 year), perimenopausal (1–5 years) or postmenopausal; hormone replacement therapy (HRT) status (women only (2HE)) was categorised as current, former or never users. The use of statins and steroids at 2HE were categorised as yes or no. Smoking status (derived from HLQ2) (never, former, current) was derived from yes and no responses to the following questions “Have you ever smoked as much as one cigarette a day for as long as a year?” and “Do you smoke cigarettes now?”. Self-reported physical activity (derived from HLQ1) was assessed using both occupational and leisure activities and individuals were assigned to one of four categories: inactive, moderately inactive, moderately active and active [51,52]. Occupational social class at 1HE was defined according to the Registrar General’s classification. Nonmanual occupations were represented by codes I, (professional) II, (managerial and technical), and IIIa (nonmanual skilled) occupations while manual occupations were represented by codes IIIb (manual skilled), IV (partly skilled) and V (unskilled) occupations [53]. 

### 2.6. Statistical Analysis

All analyses were stratified by sex as significant differences in body composition, SMM and age-related changes in bone existing [44] between men and women. *p* < 0.05 was considered to be statistically significant in individual analyses. To minimise missing data exclusions, some missing values were recoded as follows: missing menopausal status data (2.8%) as premenopausal if age <50 years and never-user of HRT, or as postmenopausal if age >55 years or a current or former HRT user. Participants missing data for other variables in the multivariable model were excluded. Participants were excluded from analyses if they had missing or extreme BIA impedance values (<300 or >1000 ohms [54]), FFM < 25, or for participants with extremes of BMI (<14 or ≥36 kg/m^2^), since bioelectrical impedance measures are considered unreliable at these levels [55]. 

### 2.7. Cross-sectional Analyses

Cross-sectional analyses were carried out using data from the 2HE, using dietary or plasma data from the 1HE; 14,179 participants had complete data for diet and muscle analyses, and 4283 had complete data for plasma and muscle analyses; the figures for BUA analyses are 14,694 and 4457, respectively (Figure 1). 

Multivariable adjusted regression with ANCOVA was used to investigate differences in indices of SMM and calcaneal BUA across sex-specific dietary intake quintiles of vitamin E intake (mg α-tocopherol equivalents). Trend testing was achieved by treating the median values for quintiles as a continuous variable [56]. Each model was adjusted for important physiological, lifestyle, and dietary factors, known to influence risk in this population. For SMM, these included age, smoking status, physical activity, social class, energy intake, percentage energy from protein, corticosteroid and statin use, menopausal and HRT status in women; for BUA, these included age, BMI, family history of osteoporosis, menopausal and HRT status in women, corticosteroid use, smoking status, physical activity, Ca intake, total energy intake, and Ca- and vitamin D-containing supplement use. The data were also analysed to take the amount of vitamin E from supplements into consideration, as excluding supplements may underestimate total nutrient intake [57]. In separate analyses, indices of SMM and calcaneal BUA were investigated across sex-specific plasma concentration quintiles of α-tocopherol, γ-tocopherol and the ratio of α:γ-tocopherol, with the covariates described above, but excluding dietary intake data. 

### 2.8. Longitudinal Analyses

Longitudinal analyses used data from the 1HE together with incident hospital-recorded fractures for the participants (all hip, spine and wrist fracture cases up to 31 March 2018); the mean follow-up time was 18.5 years (467,077 total person years), and was calculated as the time between an individual’s 1HE and this cut-off date, or death if earlier. Data for diet and fracture analyses were available for 25,223 participants; data for plasma and fracture analyses were available for 7291 participants (Figure 1). Prentice-weighted Cox regression was used to investigate associations between incidence of fractures and sex-specific quintiles of dietary vitamin E intake (mg α-tocopherol equivalents), or plasma concentrations, using the same adjustments as for the BUA models. Missing values were treated in the same way as in the BUA models. Total risk for hip, spine or wrist fracture was calculated as the risk for the first occurrence of one of these fractures; this does not consider multiple fractures, and therefore the sum of the specific-site fracture incidences does not sum to the total.

## 3. Results

### 3.1. Characteristics of the Study Population 

Selected characteristics are summarised in Table 1, stratified by dietary analysis group and sex. Mean dietary and supplement-derived intakes of α-tocopherol equivalents (mg/day) are shown for the different study groups. In the dietary model analyses, numbers of men and women are similar for the SMM and BUA measures and intakes of dietary and supplement α-tocopherol equivalents are also similar. Dietary intakes of α-tocopherol equivalents are slightly lower in the fracture dietary analyses groups, as is the percentage of participants taking vitamin E-containing supplements and the amount of α-tocopherol equivalents obtained from these supplements. No UK Reference Nutrient Intake value [58] has been defined for vitamin E, although safe intakes of α-tocopherol equivalents have been set at 4 mg for men and 3 mg for women. Of the 11,427 men in the fracture dietary analysis group, 1.7% had an intake <4 mg (*n* = 194); 1.0% of the 13,796 women had an intake <3 mg (*n* = 132). However, the European Food Safety Agency (EFSA) Panel on Dietetic Products, Nutrition and Allergies (NDA) set Adequate Intakes (AIs) of α-tocopherol for adults as 13 mg/day for men and 11 mg/d for women [59]; it was felt that Average Requirements (ARs) and Population Reference Intakes (PRIs) could not be set for α-tocopherol. Only 30.7% of the men and 25.8% of the women in the fracture cohort met these AIs. The fracture dietary analyses groups had a higher percentage of current smokers, manual workers and physically inactive participants. 

Table 2 presents selected characteristics, stratified by plasma analysis group and sex. Mean dietary and supplement-derived intakes of α-tocopherol equivalents (mg/day) are shown for the different study groups, in addition to plasma tocopherol concentrations, unadjusted and adjusted for cholesterol. In the plasma model analyses, numbers of men and women are similar for the SMM and BUA measures and concentrations of the plasma tocopherols and intakes of dietary and supplement α-tocopherol equivalents are also similar. Concentrations of the cholesterol-adjusted plasma tocopherols are also similar in the fracture plasma analysis groups, although the ratio is slightly lower in both men and women. A plasma tocopherol concentration of at least 11.6 μmol/L, or a minimum tocopherol:cholesterol ratio of 2.25 μmol/mmol is considered to be the lowest satisfactory value; the dietary requirement of vitamin E is that which is necessary to keep the ratio above this level [58], although in a recent publication by EFSA, it was considered that they were insufficient data on markers of α-tocopherol intake/status/function (e.g., plasma/serum α-tocopherol concentration, markers of oxidative damage) to calculate the requirement for α-tocopherol [59]. Of the participants in the SMM plasma analysis group, only 4 men and 8 women had values <11.6 μmol/L; one of these woman had a tocopherol:cholesterol ratio <2.25 μmol/mmol. Eight men and 13 women had tocopherol:cholesterol ratios <2.25 μmol/mmol. Once again, the larger fracture plasma analysis groups had a higher percentage of current smokers, manual workers and physically inactive participants.

### 3.2. Food Sources of α- and γ-Tocopherols

Good sources of vitamin E include plant oils—such as rapeseed, sunflower, soya, corn and olive oil—nuts, seeds and wheatgerm. The main sources of γ-tocopherol include oils [60] (especially soybean and corn oils, which are used extensively in processed foods), nuts and seeds [61] (especially walnuts, pecans and pistachios, as well as sesame, flax and pumpkin seeds), as well as spinach, carrots, avocado, dark green leafy vegetables and wheatgerm. Figure 2 shows the main food group sources for men and women. Generally, the main food groups contributing to vitamin E intake in men and women were similar, with butters, spreads and margarines being the main contributors. Foods in the grains and cereal-based products groups include both sweet and savoury biscuits, cakes, pies and quiches.

### 3.3. Correlations between Dietary Vitamin E Intake and Plasma Concentrations

A number of weak but significant correlations were found between the dietary intake of α-tocopherol equivalents and plasma concentrations of α-tocopherol. Dietary intake of α-tocopherol equivalents was significantly correlated with plasma concentration of α-tocopherol in the SMM cohort in men (*r* = 0.079, *p* < 0.001, *n* = 2232), but not in women (*r* = 0.038, *p* = 0.084, *n* = 2051). In the BUA cohort, significant correlations were found in both men (*r* = 0.083, *p* < 0.001, *n* = 2300) and women (*r* = 0.044, *p* < 0.05, *n* = 2143). In the fracture cohort, dietary intake of α-tocopherol equivalents was significantly correlated with plasma concentration of α-tocopherol in both men (*r* = 0.105, *p* < 0.001, *n* = 3707) and women (*r* = 0.08, *p* < 0.001, *n* = 3551). 

When the plasma concentration of α-tocopherol was adjusted for total cholesterol, the correlations with dietary intake were found to be slightly stronger. Dietary intake of α-tocopherol equivalents was significantly correlated with plasma concentration of α-tocopherol in the SMM cohort in both men (*r* = 0.136, *p* < 0.001, *n* = 2232) and women (*r* = 0.1203, *p* < 0.001, *n* = 2051). In the BUA cohort, significant correlations were found in both men (*r* = 0.131, *p* < 0.001, *n* = 2300) and women (*r* = 0.125, *p* < 0.001, *n* = 2143). In the fracture cohort, dietary intake of α-tocopherol equivalents was significantly correlated with plasma concentration of α-tocopherol in both men (*r* = 0.1565, *p* < 0.001, *n* = 3707) and women (*r* = 0.153, *p* < 0.001, *n* = 3551). 

### 3.4. Associations between Dietary Vitamin E Intakes and Indices of SMM

Significant positive associations were found between sex-specific quintiles of dietary vitamin E and FFM and FFM_BMI_ (*p*-trend < 0.001 in both men and women), after adjustments for covariates (Table 3), with significant interquintile differences (Q5 versus Q1) in FFM of +1.0% (*p* < 0.001) in both men, and women, and in FFM_BMI_ of +1.7% (*p* < 0.001) in men and +1.9% (*p* < 0.001) in women. The addition of the amount of vitamin E derived from supplements to the fully adjusted models did not alter the associations.

### 3.5. Associations between Plasma Vitamin E Concentrations and Indices of SMM

In general, similar linear trends were found for both men and women, with those across quintiles of α- and γ-tocopherol tending to be in the same direction, and the trend for the ratio of α-tocopherol: γ-tocopherol was in the opposite direction (Table 4). Linear trends in both men and women were most apparent across quintiles of plasma γ-tocopherol. In adjusted plasma model analyses, there was a significant positive trend across both α- and γ-tocopherol quintiles for FFM in men (*p* < 0.001) and women (*p* < 0.01). However, for FFM, a significant negative trend was found across quintiles of the ratio of α-tocopherol:γ-tocopherol in both men and women (*p* < 0.001). Whereas, across quintiles of the ratio of α-tocopherol: γ-tocopherol, significant positive trends were found for FFM in both men (*p* < 0.001) and women (*p* < 0.01), and significant negative trends for FFM_BMI_ in both men and women (*p* < 0.001). In the adjusted model for FFM, significant differences were found between quintile 1 and quintiles 4 and 5 of plasma γ-tocopherol in women (*p* < 0.01 and <0.05, respectively), whereas in men, significant differences were found between quintile 1 and quintiles 2 (*p* < 0.01), 4 (*p* < 0.05) and 5 (*p* < 0.01) of the ratio of α-tocopherol:γ-tocopherol. In the adjusted model for FFM_BMI_, significant differences were found between Q1 and Q3 of plasma α-tocopherol in women (*p* < 0.05). Regarding plasma γ-tocopherol, significant differences were found between Q1 and Q5 of plasma in men (*p* < 0.01) and between Q1 and Q4 (*p* < 0.05) and 5 (*p* < 0.01) in women. In women, significant differences were found between Q1 and Q4 (*p* < 0.01) and Q5 (*p* < 0.05) of the ratio of α-tocopherol:γ-tocopherol.

### 3.6. Associations between Dietary Vitamin E Intakes and Bone Density Status

Mean calcaneal BUA values, stratified by sex and quintiles of dietary vitamin E, are shown in Table 5, for unadjusted data and the fully adjusted model. Significant positive associations across quintiles of dietary vitamin E intake were evident in both men and women, after adjustments for covariates (*p*-trend < 0.001 in both men and women). In the fully adjusted model, a significant difference was identified in men, for Q3 versus Q1 (+1.8%; *p* < 0.05). Further adjustment for the amount of vitamin E derived from supplements did not modify the associations.

### 3.7. Associations between Plasma Vitamin E Concentrations and Bone Density Status 

Analysis of mean calcaneal bone density measures, stratified by sex and quintiles of plasma vitamin E concentrations, is shown in Table 6, for both the unadjusted and fully adjusted models. In both men and women, mean BUA measures tended to significantly increase across quintiles of plasma α- and γ-tocopherol and decrease across quintiles of the ratio of α-tocopherol:γ-tocopherol (*p* < 0.001). No significant differences were found between quintile 1 and any of the other quintiles in the adjusted models, in either men or women. 

### 3.8. Associations between Dietary Vitamin E Intakes and Fracture Risk

In the fully adjusted dietary model analyses, significant positive associations were evident between quintiles of vitamin E intake and risk for total fractures and hip fractures in both men and women (*p* < 0.001), but negative associations were found for wrist fractures in women (*p* < 0.01) (Table 7). In men, both total fracture risk and wrist fracture risk were significantly lower in Q2 versus Q1 (0.79; 95% CI 0.64, 0.98; *p* < 0.05 and 0.51; 95% CI 0.29, 0.90; *p* < 0.05, respectively). In women, hip fracture risk was significantly lower in Q2 versus Q1 (0.81; 95% CI 0.66, 0.99; *p* < 0.05). The addition of supplement-derived vitamin E intake to the fully adjusted models did not alter the associations.

### 3.9. Associations between Plasma Vitamin E Concentrations and Fracture Risk

In the fully adjusted plasma vitamin E analyses, significant linear trends were found for risk of total and hip fractures and quintiles of plasma α-, γ-tocopherol and the ratio of α-tocopherol:γ-tocopherol in men (*p* < 0.05) (Table 8). The risk for both total and hip fracture decreased across quintiles of plasma α-tocopherol and the ratio of α-tocopherol:γ-tocopherol, but increased across quintiles of γ-tocopherol for hip fracture risk and tended to decrease for total fracture risk. In women, in the fully adjusted plasma vitamin E analyses, significant linear trends were found for risk of total and hip fractures and quintiles of plasma α-, γ-tocopherol and the ratio of α-tocopherol:γ-tocopherol (*p* < 0.05). The risk for total and hip fractures tended to decrease across quintiles of plasma α-tocopherol and the ratio of α-tocopherol:γ-tocopherol but increase across the quintiles of γ-tocopherol. In men, spine fracture risk was significantly higher in Q3 versus Q1 (2.00; 95% CI 1.00, 4.00; *p* < 0.05) in the fully adjusted ratio of α-tocopherol:γ-tocopherol model. Total fracture risk in women was significantly higher in Q4 versus Q1 (1.29; 95% CI 1.01, 1.65; *p* < 0.05) in the fully adjusted plasma γ-tocopherol model. However, hip fracture risk in women was significantly lower in Q3 versus Q1 (0.61; 95% CI 0.41, 0.89; *p* < 0.05) in the fully adjusted ratio of α-tocopherol:γ-tocopherol model. Wrist fracture risk in women was significantly lower in Q5 versus Q1 (0.56; 95% CI 0.32, 0.98; *p* < 0.05) in the fully adjusted plasma α-tocopherol model.

The associations of dietary intake and plasma concentrations of vitamin E with SMM, BUA and fracture risk are summarised in Table 9.

## 4. Discussion

To our knowledge, this is the first study using data from a large population cohort of middle- and older-aged men and women in the UK to assess the associations between both dietary vitamin E intake and plasma vitamin E concentrations and indices of SMM, bone density status and fracture risk. Our results show significant positive associations between both dietary vitamin E intake and plasma concentrations of both serum cholesterol-adjusted α- and γ-tocopherol and FFM and BUA, and generally significant positive associations for fracture risk. The associations found with vitamin E for bone density status and fracture risk are independent of vitamin D and calcium intake, which are both known to be relevant for bone health. The results from this study indicate protection for musculoskeletal health with higher intakes and blood concentrations of vitamin E.

In the EPIC-Norfolk study, the mean daily dietary intakes of vitamin E in 11,535 men and 13,972 women, assessed using a 7dDD and expressed in milligrams of α-tocopherol equivalents, were 11.62 (SD 5.24) and 9.27 (SD 3.78) mg [48] which is very similar to the mean daily intakes in the cohorts in these analyses. These dietary intakes are in agreement with those from other European countries [62,63]. Although in relation to the AIs defined by EFSA [59], only 31% of men and 26% of women met the AIs.

Most studies conducted in the US reported higher average plasma γ-tocopherol levels as compared to studies conducted in Europe [32]. This is likely explained by the fact that γ-tocopherol is the major form (approximately 70%) of vitamin E in the US diet [31]. Therefore, this study focuses on European comparisons. The mean plasma α- and γ-tocopherol concentrations in our study cohorts are similar to those found in a healthy Irish adult population [63] and in the UK National Diet and Nutrition Survey (NDNS) [64]. More than 99% of both men and women in our study cohorts had higher plasma tocopherol concentrations or tocopherol:cholesterol ratios than the minimum satisfactory values [58].

Our study found a number of weak but significant correlations between the dietary intake of α-tocopherol equivalents and plasma concentrations of α-tocopherol, which has also been observed in a small German study [65] (*n* = 92; *r* = 0.14) but no associations have been found in a number of other European studies [66,67,68]. However, Kardinaal et al. found a significant age- and sex-adjusted correlation *(r* = 0.24, *p* < 0.05) for α-tocopherol between intake and adipose tissue levels in a small study of healthy adults [66], whereas no correlation was found between the adipose tissue level of alpha-tocopherol and dietary intake by Andersen et al. [67].

Significant positive trends in FFM and FFM_BMI_ were evident across increasing quintiles of dietary vitamin E intake for both sexes, after adjusting for important covariates. Similar linear trends were generally apparent in both men and women for plasma vitamin E concentrations, with those across quintiles of α- and γ-tocopherol tending to be in the same direction (positive for FFM but negative for FFM_BMI_), and the trend for the ratio of α-tocopherol:γ-tocopherol in the opposite direction (negative for FFM but positive for FFM_BMI_). These seemingly contradictory findings are not surprising as the increase in the ratio across quintiles is due to decreasing plasma α- but increasing γ-tocopherol concentrations. Whereas across increasing quintiles of plasma α-tocopherol, γ-tocopherol also increased, and across increasing quintiles of γ-tocopherol, plasma concentrations of α-tocopherol tended to decrease slightly. To date, most studies that have investigated the potential role of vitamin E in muscle health have focused on muscle function and strength rather than muscle mass and found that higher dietary vitamin E intakes [22,26] and plasma vitamin E concentrations were associated with higher strength measures and physical performance tests or lower levels of frailty [21,23,24,69,70,71]. Findings from our study support the importance of vitamin E to skeletal muscle health.

Significant trends were apparent for BUA across quintiles of dietary vitamin E intake and plasma concentrations in both men and women, with BUA tending to increase across quintiles of dietary vitamin E intake and plasma α- and γ-tocopherol but decrease across quintiles of the ratio of α-tocopherol:γ-tocopherol (*p* < 0.001). Significant positive associations were evident for dietary vitamin E intake and risk for total and hip fractures in both men and women (*p* < 0.001), but a significant negative association was found for wrist fracture risk in women (*p* < 0.01), where the greatest number of fractures was found in Q1, and the lowest in Q5; it is possible that this association may be artefactual. In plasma vitamin E analyses, significant linear trends were found for total fracture risk in men (*p* < 0.05) with the risk of total fractures generally decreasing across the quintiles in all 3 models. Significant linear associations were also found for hip fracture risk in men (*p* < 0.01), with the risk of hip fracture decreasing across quintiles of plasma α-tocopherol and the ratio of α-tocopherol:γ-tocopherol but increasing across quintiles of γ-tocopherol. In women, significant linear trends were found for risk of total and hip fractures and plasma vitamin E (*p* < 0.05), with the risk of fractures generally decreasing across quintiles of plasma α-tocopherol and the ratio of α-tocopherol:γ-tocopherol but increasing across quintiles of γ-tocopherol.

In the Aberdeen Prospective Osteoporosis Screening Study (APOSS) cohort, no biologically meaningful changes in BMD or bone resorption and formation markers with dietary intakes or serum concentrations of tocopherols or the α/γ ratio were found in perimenopausal and postmenopausal women [72], although dietary vitamin E intake was negatively associated with femoral neck BMD in early postmenopausal women in Scotland [73]. A recent study of nutrient intake and BMD in postmenopausal women found that a high intake of vitamin E had a negative effect on BMD [74]. Low serum concentrations of α- tocopherol have been associated with an increased risk of hip fracture in elderly men and women [75] and an increased osteoporosis risk in postmenopausal women [27]. Both low intakes and serum concentrations of α-tocopherol were associated with an increased rate of fracture in elderly Swedish men and women [28]. There are a number of plausible explanations for the heterogeneity of the conclusions of the aforementioned epidemiological studies—the use of different covariates in the multivariable models, inconsistent measurement validity of biomarkers and the application of various exclusion criteria regarding the study sample—although most concur with our findings. Whether or not these study findings have any biological significance is unclear.

A recent review on the beneficial and detrimental effects of oxidative stress on human health concluded that α- and γ-tocopherol forms of vitamin E exert a differential set of biological effects, which cannot always be regarded as positive to human health [76]. Recent data have also suggested that plasma α-tocopherol concentrations are more dependent on mechanisms that control circulating lipids rather than those related to its absorption and initial incorporation into plasma [77]; α-tocopherol was found to remain in circulation longer in participants with higher serum lipids, but its absorption was not dependent on the plasma lipid status.

In contrast to a high affinity to α-tocopherol (100%), α-tocopherol transfer protein (α-TTP) has a much lower affinity towards other vitamin E forms; 50%, 10–30%, and 1% affinity to β-tocopherol, γ-tocopherol, and δ-tocopherol, respectively [78], and plays an important role in the maintenance of high concentrations of α-tocopherol in plasma and some tissues [79,80]. A reduction in plasma γ-tocopherol during enhanced intake of α-tocopherol, such as through supplemental intake, can be explained by the more rapid metabolism of γ-tocopherol occurring when α-tocopherol intake is increased [81]. Chylomicron-associated tissue uptake of vitamin E may contribute to the accumulation of non-α-tocopherol forms of vitamin E such as γ-tocopherol in human skin, adipose tissue, and muscle, where unexpectedly high concentrations of γ-tocopherol were observed, in contrast to its low levels in the plasma [82]. Many unique properties have been attributed to γ-tocopherol and its metabolites [83], which exhibit sometimes enhanced or different activities of α-tocopherol such as natriuretic, anti-inflammatory, antitumoural activities, as summarised in a recent review [84]. The ratio of α-tocopherol:γ-tocopherol is suggested as a correction method as it would respond to even a small increase in α-tocopherol from supplementation that may not be clearly evident in plasma α-tocopherol concentrations [85]. Findings from the analyses in this study have shown that adjustment for the amount of vitamin E from supplements did not affect the associations.

The interactions between these two tocopherols are complex within the body and it must also be remembered that the bioavailability of vitamin E is influenced by a number of factors, including other nutrients, genetics, absorption, transport and metabolism [86]. With regard to other nutrients and food intake, data from the NDNS found that α-tocopherol correlated directly with “healthy” nutrient choices (intrinsic sugars, dietary fibre, and vitamins) and inversely with “unhealthy” choices (extrinsic sugars and monounsaturated fats—i.e., avoidance of polyunsaturated fat), whilst γ-tocopherol and the γ-tocopherol:α-tocopherol ratio related inversely with “healthy” choices, with the authors concluding that the γ-tocopherol:α-tocopherol ratio may reveal poor dietary choices, which may subsequently lead to health issues in later life [87].

A number of possible mechanisms have been suggested illustrating how vitamin E may slow down the aging of skeletal muscle: (1) by improving antioxidant capacity, thereby reducing oxidative stress and inflammation; (2) improving membrane repair and increasing survival of damaged skeletal muscle by reducing oxidized phospholipid formation; (3) improving mitochondrial efficiency; (4) decreasing glycogen usage in skeletal muscle, while increasing fat metabolism; (5) enhancing muscle regeneration capacity; (6) stabilize insulin structure and improve insulin sensitivity of skeletal muscle [35]. It is thought that reduction in oxidative stress may also be a plausible mechanism whereby vitamin E protects bone, although the reported opposing inflammatory functions of α- and γ-tocopherol may result in an increase in bone resorption without affecting bone formation [36]. However, further research is needed to investigate the potential effects of other tocopherols and tocotrienols on sarcopenic and osteoporotic risk factors. 

The strengths of our study include a large population size of middle-aged and elderly men and women, from whom we had measures of dietary and supplemental intake, obtained from 7dDDs, in addition to plasma concentrations of vitamin E (α- and γ-tocopherol), in order to study the potential associations of vitamin E with indices of SMM, BUA and fracture risk (over 18.5 years of follow-up). Limitations of our study include the observational and cross-sectional study design regarding SMM and BUA measurements, precluding us from inferring causation, and the use of self-reported measures for dietary intake and physical activity. However, the prospective nature of our study of fracture risk and long follow-up for end points of 18.5 years are advantages. In addition, the 7dDDs developed for use in the EPIC-Norfolk study have previously been validated and are expected to produce a more precise measure of dietary intake than 24 h diet recalls or food frequency questionnaires [48]. Plasma vitamin E concentrations were only available for a small subset of the cohort, which may have reduced the power of our analyses. Nevertheless, the availability of plasma concentrations, which are not subject to nonrandom biases that can affect questionnaire-based measurements, is a strength of our study, although these concentrations may be affected by various physiological effects. SMMs were calculated from weight, height and bioelectrical impedance measurements, and not from potentially more accurate and precise methods, such as DEXA, computer tomography or magnetic resonance imaging; however, this method has comparable acceptability in population studies [88]. The dietary and lifestyle data, including the consumption of corticosteroids, HRT and dietary supplements, used in the longitudinal analyses were collected at 1HE and we were unable to account for any changes in exposures which may have occurred over time and potentially affected the associations.

## 5. Conclusions

Our research has found significant positive associations between greater intakes of dietary vitamin E and SMM indices, bone density status and total and hip fracture risk in both middle-aged and elderly men and women, with the scale of effects ranging from 0.88% to 1.91% (*p* < 0.001). Associations found with circulating plasma α- and γ-tocopherol generally agreed with the dietary data. These findings suggest that dietary vitamin E intake may play a role in musculoskeletal health and provides evidence of the benefits of higher vitamin E intakes, similar to the AIs of α-tocopherol for adults of 13 mg/day for men and 11 mg/d for women, as recommended by EFSA. These intakes can be achieved by eating a varied and balanced diet, including the consumption of foods rich in vitamin E, such as oily seeds and their derivatives, nuts and cereals rich in vitamin E, including fortified breakfast cereals. Where it is not possible to obtain adequate intakes through the diet, vitamin E supplements should be consumed, especially in those at sarcopenic or osteoporotic risk. Further investigation is required to understand the relationships with plasma concentrations and musculoskeletal health.

## Figures and Tables

**Figure 1 antioxidants-10-00159-f001:**
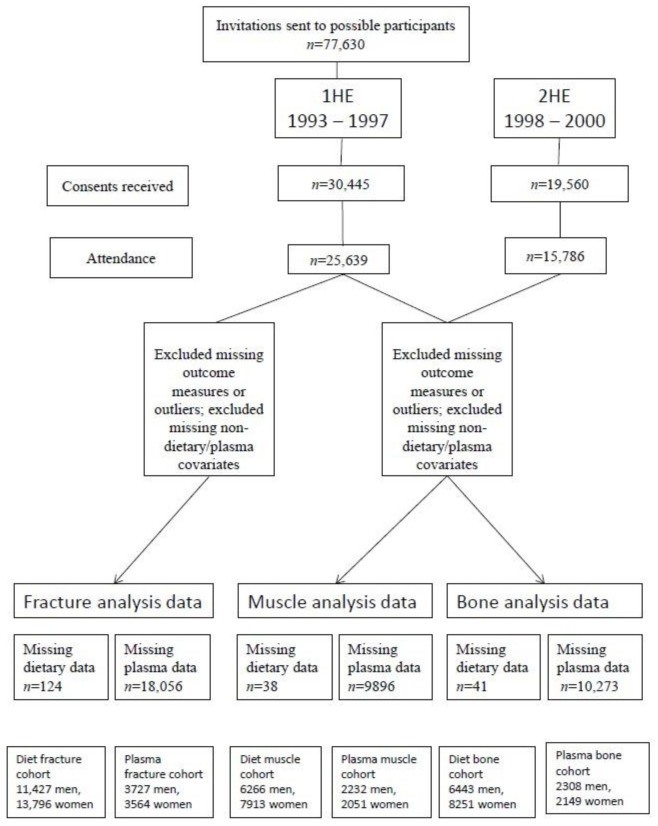
Flowchart of participants included in the analyses.

**Figure 2 antioxidants-10-00159-f002:**
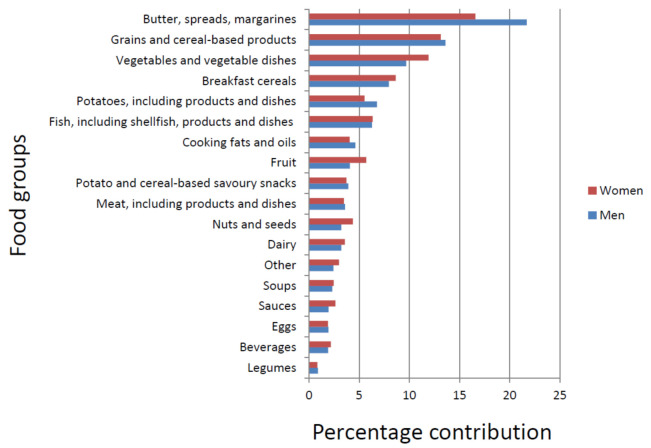
Percentage composition of food groups for vitamin E intake of men and women in the EPIC-Norfolk cohort. “Butters, spreads and margarines” include reduced fat types. “Grains and cereal-based products” include rice and rice-based dishes, pasta, sweet and savoury flans, pies and quiches, biscuits, cakes, breads and bread rolls. “Vegetables and vegetable-based dishes” include raw and cooked vegetables, vegetable dishes and mixed salads. “Breakfast cereals” include porridge and muesli. “Potatoes, including products and dishes” include potatoes, potato products, dishes and salads. “Fish, including shellfish, products and dishes” includes shellfish and fish-based dishes. “Cooking fats and oils” include hard margarines, animal fats, vegetable fats and ghee. “Fruit” includes fresh, cooked and canned fruit. “Potato and cereal-based savoury snacks” include crisps and other potato-based snacks and bread/pastry type snacks. “Meat, including products and dishes” include unprocessed white and red meats (and products and dishes), processed meat products and offal, including products and dishes. “Nuts and seeds” include nuts, seeds and nut butters. “Dairy” includes milk, cheese, yoghurts and dairy-based desserts. “Other” includes sugar, herbs and spices and dietetic products. “Soups” include homemade, canned and reconstituted dried soups. “Sauces” include sweet and savoury sauces, gravies, salad dressings, pickles, chutneys and stuffing. “Eggs” include eggs and sweet and savoury egg dishes. “Beverages” include alcoholic and nonalcoholic drinks, such as tea, coffee, water, soft drinks, fruit juice and squashes. “Legumes” include beans, pulses and lentils, dried, cooked and canned, and legume-based salads.

**Table 1 antioxidants-10-00159-t001:** Selected characteristics of the European Prospective Investigation Into Cancer and Nutrition (EPIC)-Norfolk cohort population, stratified by sex and diet analysis group.

	Diet Analysis Cohort *				Diet Analysis Cohort *				Diet Analysis Cohort ^†^			
	(SMM)				(BUA)				(Fractures)			
	Men (*n* = 6266)		Women (*n* = 7913)		Men (*n* = 6443)		Women (*n* = 8251)		Men (*n* = 11,427)		Women (*n* = 13,796)	
Age (years)	62.8	9.0	61.5	9.0	62.9	9.0	61.5	9.0	59.6	9.3	58.9	9.3
BMI (kg/m^2^)	26.7	3.0	26.1	3.7	26.9	3.3	26.5	4.4	26.5	3.3	26.2	4.4
Dietary α-tocopherol equivalents intake (mg/day)	11.90	5.00	9.52	3.73	11.91	5.04	9.50	3.74	11.63	5.24	9.29	3.78
Energy intake (kcal/day)	2287	501	1736	378	2285	502	1732	380	2240	528	1695	395
Protein intake (% energy)	14.8	2.4	15.5	2.8								
Calcium intake (mg/day)					942	289	785	243	919	297	767	248
Vitamin D intake (µg/day)					3.88	2.82	2.99	1.94	3.73	2.75	2.94	2.08
Vitamin E-containing supplement use, n (%)	1000	16.0	2077	26.2	1024	15.9	2153	26.1	1587	13.9	3244	23.5
Supplemental α-tocopherol equivalents (mg/day)	37.27	91.86	35.99	77.52	37.73	93.61	35.12	75.72	35.10	85.16	35.29	78.73
Vitamin D-containing supplement use, n (%)					1612	25.0	2752	33.4	2551	22.3	4232	30.7
Supplemental vitamin D (μg/day)					4.34	2.84	4.17	2.57	4.25	2.75	4.18	2.66
Calcium-containing supplement use, n (%)					102	1.6	504	6.1	165	1.4	742	5.4
Supplemental calcium (mg/day)					200	195	347	269	196	190	342	268
FFM (kg)	61.6	5.8	40.6	4.5								
FFM_BMI_	2.32	0.26	1.59	0.26								
BUA (dB/MHz)					90.06	17.51	72.09	16.46				
Total incident fractures, n (%)									877	7.7	2092	15.2
Incident hip fractures, n (%)									356	3.1	971	7.0
Incident spine fractures, n (%)									223	2.0	357	2.6
Incident wrist fractures, n (%)									155	1.4	504	3.6
Social class, n (%)												
Professional	516	8.2	544	6.9	523	8.12	565	6.85	860	7.5	863	6.3
Managerial	2566	41.0	2919	36.9	2630	40.82	3016	36.55	4299	37.6	4722	34.2
Skilled nonmanual	782	12.5	1545	19.5	804	12.48	1600	19.39	1404	12.3	2684	19.4
Skilled manual	1401	22.4	1565	19.8	1440	22.35	1636	19.83	2840	24.8	2847	20.6
Semiskilled	771	12.3	937	11.8	803	12.46	1001	12.13	1501	13.1	1805	13.1
Nonskilled	145	2.3	262	3.3	153	2.37	282	3.42	334	2.9	536	3.9
Missing	85	1.4	141	1.8	90	1.4	151	1.83	189	1.6	339	2.5
Smoking status, n (%)												
Current	490	7.8	638	8.1	509	7.9	660	8.0	1391	12.2	1560	11.3
Former	3495	55.8	2547	32.2	3609	56.01	2697	32.69	6232	54.5	4446	32.2
Never	2281	36.4	4728	59.8	2325	36.09	4894	59.31	3804	33.3	7790	56.5
Physical activity, n (%)												
Inactive	1712	27.3	2043	25.8	1779	27.61	2165	26.24	3516	30.8	4174	30.3
Moderately inactive	1576	25.2	2575	32.5	1615	25.07	2695	32.66	2810	24.6	4429	32.1
Moderately active	1567	25.0	1920	24.3	1601	24.85	1980	24.00	2635	23.1	3074	22.3
Active	1411	22.5	1375	17.4	1448	22.47	1411	17.10	2466	21.6	2119	15.4
Family history of osteoporosis, n (%)												
No	6098	97.4	7413	93.7	6267	97.3	7731	93.7	11,121	97.3	12,988	94.1
Yes	166	2.6	500	6.3	176	2.7	520	6.3	306	2.7	808	5.9
Corticosteroid use, n (%)												
Current or former (>3 months)	260	4.2	402	5.1	270	4.2	422	5.1	349	3.0	478	3.5
Never (<3 months)	6006	95.8	7511	94.9	6173	95.8	7829	94.9	11,078	97.0	13,318	96.5
Statin use, n (%)												
No	5922	94.5	7623	96.3								
Yes	344	5.5	290	3.7								
Menopausal status, n (%)												
Premenopausal			472	6.0			482	5.8			233	16.9
Perimenopausal (<1 y)			265	3.4			272	3.3			750	5.4
Perimenopausal (1–5 y)			1389	17.6			1455	17.6			2473	17.9
Postmenopausal			5787	73.1			6042	73.2			8239	59.7
HRT, n (%)												
Current			1693	21.4			1757	21.3			2802	20.3
Former			1417	17.9			1482	18.0			1570	11.4
Never			4803	60.7			5012	60.7			9424	68.3

Values are mean ± SD or frequency (percentage). SMM = skeletal muscle mass; BUA = broadband ultrasound attenuation; FFM = fat-free mass; HRT = hormone replacement therapy. * SMM and BUA group characteristics at second health examination (2HE) (time of ultrasound), unless only available at 1HE. ^†^ Fracture group characteristics at 1HE or time of consent. SMM: men = 6264 for family history of osteoporosis.

**Table 2 antioxidants-10-00159-t002:** Selected characteristics of the EPIC-Norfolk cohort population, stratified by sex and plasma analysis group.

	Serum Analysis—SMM *				Serum Analysis—BUA *				Serum Analysis—Fractures ^†^			
	Men (*n* = 2232)		Women (*n* = 2051)		Men (*n* = 2308)		Women (*n* = 2149)		Men (*n* = 3727)		Women (*n* = 3564)	
Age (years)	67.0	7.5	64.5	8.5	67.0	7.5	64.5	8.5	64.39	7.87	62.02	8.72
BMI (kg/m^2^)	26.8	3.0	26.3	3.7	27.0	3.4	26.8	4.3	26.74	3.33	26.39	4.20
Dietary α-tocopherol equivalents intake (mg/day)	11.35	4.70	9.30	3.56	11.38	4.72	9.28	3.59	11.02	4.76	9.12	3.69
Serum α-tocopherol (μmol/L)	26.44	7.74	28.39	8.35	26.47	7.88	28.42	8.34	26.36	7.90	28.53	8.50
Serum chol-adjusted α-tocopherol (μmol/mmol)	4.35	1.06	4.43	1.05	4.37	1.08	4.43	1.05	4.34	1.07	4.42	1.06
Serum γ-tocopherol (μmol/L)	1.83	0.90	1.83	0.91	1.83	0.90	1.84	0.93	1.86	0.90	1.88	0.94
Serum chol-adjusted γ-tocopherol (μmol/mmol)	0.30	0.14	0.28	0.13	0.30	0.14	0.29	0.13	0.31	0.14	0.29	0.14
Serum α-tocopherol:γ-tocopherol ratio (chol-adjusted)	17.97	16.27	20.61	29.31	17.96	16.53	20.41	28.73	17.55	15.29	19.92	25.35
FFM (kg)	61.31	5.80	40.45	4.66								
FFM_BMI_	2.30	0.25	1.56	0.25								
BUA (dB/MHz)					89.45	17.75	69.83	16.16				
Total incident fractures, n (%)									308	8.3	643	18.0
Incident hip fractures, n (%)									129	3.5	313	8.8
Incident spine fractures, n (%)									89	2.4	124	3.5
Incident wrist fractures, n (%)									40	1.1	152	4.3
Energy intake (kcal/day)	2217	488	1715	374	2218	489	1709	377	2161	502	1675	383
Calcium intake (mg/day)					924	283	780	250	900	283	764	249
Vitamin D intake (µg/day)					3.95	2.72	3.04	1.97	3.85	2.71	3.02	2.21
Vitamin E-containing supplement use, n (%)	350	15.7	512	25.0	360	15.6	534	24.8	503	13.5	806	22.6
Supplemental α-tocopherol equivalents (mg/day)	36.05	92.50	39.84	83.53	36.13	92.35	39.12	82.00	34.54	85.44	38.10	80.96
Vitamin D-containing supplement use, n (%)	596	26.7	657	32.0	607	26.3	682	31.7	893	24.0	1075	30.2
Supplemental vitamin D (μg/day)	4.35	2.78	4.30	2.71	4.35	2.77	4.35	2.74	4.27	2.70	4.25	2.80
Calcium-containing supplement use, n (%)	38	1.7	109	5.3	38	1.6	113	5.3	53	1.4	174	4.9
Supplemental calcium (mg/day)	195	193	334	250	195	193	338	248	230	219	314	249
Social class, n (%)												
Professional	179	8.02	114	5.56	181	7.84	120	5.58	258	6.92	192	5.39
Managerial	905	40.55	731	35.64	934	40.47	762	35.46	1382	37.08	1195	33.53
Skilled nonmanual	293	13.13	422	20.58	302	13.08	436	20.29	475	12.74	708	19.87
Skilled manual	493	22.09	403	19.65	509	22.05	422	19.64	926	24.85	711	19.95
Semiskilled	280	12.54	262	12.77	294	12.74	280	13.03	504	13.52	502	14.09
Nonskilled	48	2.15	68	3.32	51	2.21	76	3.54	115	3.09	145	4.07
Missing	34	1.52	51	2.49	37	1.60	53	2.47	67	1.8	111	3.11
Smoking status, n (%)												
Current	165	7.39	145	7.07	168	7.28	148	6.89	410	11.00	388	10.89
Former	1337	59.90	668	32.57	1394	60.40	711	33.09	2240	60.10	1171	32.86
Never	730	32.71	1238	60.36	746	32.32	1290	60.03	1077	28.90	2005	56.26
Physical activity, n (%)												
Inactive	705	31.59	581	28.33	740	32.06	614	28.57	1363	36.57	1212	34.01
Moderately inactive	537	24.06	678	33.06	545	23.61	714	33.22	883	23.69	1121	31.45
Moderately active	525	23.52	485	23.65	539	23.35	502	23.36	790	21.20	769	21.58
Active	465	20.83	307	14.97	484	20.97	319	14.84	691	18.54	462	12.96
Family history of osteoporosis, n (%)												
No	2177	97.58	1929	94.05	2249	97.44	2025	94.23	3645	97.8	3381	94.87
Yes	54	2.42	122	5.95	59	2.56	124	5.77	82	2.2	183	5.13
Corticosteroid use, n (%)												
Current or former (>3 months)	123	5.51	116	5.66	130	5.63	123	5.72	150	4.02	142	3.98
Never (<3 months)	2109	94.49	1935	94.34	2178	94.37	2026	94.28	3577	95.98	3422	96.02
Menopausal status, n (%)												
Premenopausal			31	1.51			33	1.54			287	8.05
Perimenopausal (<1 y)			44	2.15			44	2.05			155	4.35
Perimenopausal (1–5 y)			268	13.07			283	13.17			534	14.98
Postmenopausal			1708	83.28			1789	83.25			2588	72.62
HRT, n (%)												
Current			396	19.31			410	19.08			660	18.52
Former			348	16.97			370	17.22			390	10.94
Never			1307	63.73			1369	63.70			2514	70.54

Values are mean ± SD or frequency (percentage). SMM = skeletal muscle mass; BUA = broadband ultrasound attenuation; FFM = fat-free mass; HRT = hormone replacement therapy. Serum BUA: Men = 2300 for vitamin E, vitamin D, calcium and energy intakes; women = 2143 for vitamin E, vitamin D, calcium and energy intakes. Serum fracture: men = 3707 for vitamin E, vitamin D, calcium and energy intakes; women = 3551 for vitamin E, vitamin D, calcium and energy intakes. SMM: men = 2231 for family history of osteoporosis; women = 2045 for body mass index (BMI). * SMM and BUA group characteristics at 2HE (time of ultrasound), unless only available at 1HE. ^†^ Fracture group characteristics at 1HE or time of consent.

**Table 3 antioxidants-10-00159-t003:** Associations between quintiles of dietary vitamin E and skeletal muscle mass in men and women aged 42–82 years.

Men (*n* = 6266)	Dietary α-Tocopherol Equivalents Intake (mg/day)		FFM		FFM_BMI_	
Quintile	Mean ± SD	Median	Unadjusted	Adjusted	Unadjusted	Adjusted
1 *(n* = 1254)	6.45 ± 1.19	6.72	60.56 ± 0.17	61.23 ± 0.16	2.25 ± 0.01	2.30 ± 0.01
2 (*n* = 1253)	9.00 ± 0.54	9.03	61.29 ± 0.16 **	61.45 ± 0.14	2.31 ± 0.01 ***	2.32 ± 0.01 *
3 (*n* = 1253)	10.93 ± 0.62	10.88	61.64 ± 0.17 ***	61.67 ± 0.14 *	2.32 ± 0.01 ***	2.32 ± 0.01 *
4 (*n* = 1253)	13.45 ± 0.90	13.36	62.20 ± 0.16 ***	61.86 ± 0.15 **	2.36 ± 0.01 ***	2.34 ± 0.01 **
5 (*n* = 1253)	19.67 ± 4.54	18.30	62.40 ± 0.16 ***	61.86 ± 0.15 **	2.38 ± 0.01 ***	2.34 ± 0.01 **
Q5–Q1 diff ^1^			1.84	0.63	0.13	0.04
% diff ^2^			3.04	1.03	5.78	1.74
*p* trend			<0.001	<0.001	<0.001	<0.001
**Women (*n* = 7913)**						
Quintile	Mean ± SD	Median	Unadjusted	Adjusted	Unadjusted	Adjusted
1 (*n* = 1583)	5.36 ± 0.99	5.58	39.95 ± 0.12	40.45 ± 0.12	1.52 ± 0.01	1.57 ± 0.01
2 (*n* = 1583)	7.41 ± 0.44	7.42	40.30 ± 0.11 *	40.47 ± 0.11	1.56 ± 0.01 ***	1.57 ± 0.01
3 (*n* = 1582)	8.94 ± 0.45	8.96	40.64 ± 0.11 ***	40.65 ± 0.11	1.59 ± 0.01 ***	1.59 ± 0.01
4 (*n* = 1583)	10.74 ± 0.64	10.70	41.04 ± 0.11 ***	40.80 ± 0.11 *	1.62 ± 0.01 ***	1.60 ± 0.01 **
5 (*n* = 1582)	15.14 ± 3.55	14.13	41.31 ± 0.11 ***	40.87 ± 0.12 *	1.64 ± 0.01 ***	1.60 ± 0.01 **
Q5–Q1 diff ^1^			1.36	0.42	0.12	0.03
% diff ^2^			3.4	1.04	7.89	1.91
*p* trend			<0.001	<0.001	<0.001	<0.001

Values are presented as means ± SEM. The *p*-trend was calculated using ANCOVA. ^1^ Q5–Q1 calculates the absolute difference between the means of quintile (Q) 5 and Q1. ^2^ % difference calculates the percentage difference between the means of Q5 and Q1. * *p* < 0.05; ** *p* < 0.01; *** *p* < 0.001 versus quintile 1, according to ANCOVA. FFM = fat-free mass. FFM and FFM_BMI_—adjusted model includes age, total energy, protein intake as a percentage of total energy, smoking status, physical activity, corticosteroid use, menopausal status, HRT use, statins use, social class and fat mass (FFM only).

**Table 4 antioxidants-10-00159-t004:** Multivariate adjusted SMM indices for EPIC-Norfolk participants, stratified by sex and quintiles of plasma α-tocopherol, plasma γ-tocopherol and α-tocopherol:γ-tocopherol ratio, adjusted for blood cholesterol measurement.

Men (*n* = 2232)	Plasma α-Tocopherol, Adjusted for Cholesterol (μmol/mmol)		FFM		FFM_BMI_	
Quintile	Mean ± SD	Median	Unadjusted	Adjusted	Unadjusted	Adjusted
1 (*n* = 447)	3.17 ± 0.36	3.25	61.03 ± 0.27	61.13 ± 0.24	2.31 ± 0.01	2.31 ± 0.01
2 (*n* = 446)	3.80 ± 0.13	3.80	61.13 ± 0.25	61.14 ± 0.24	2.31 ± 0.01	2.31 ± 0.01
3 (*n* = 447)	4.23 ± 0.13	4.23	61.76 ± 0.28	61.72 ± 0.24	2.31 ± 0.01	2.32 ± 0.01
4 (*n* = 446)	4.71 ± 0.15	4.70	61.24 ± 0.28	61.31 ± 0.24	2.31 ± 0.01	2.31 ± 0.01
5 (*n* = 446)	5.86 ± 1.14	5.47	61.36 ± 0.29	61.23 ± 0.24	2.28 ± 0.01 *	2.28 ± 0.01
Q5-Q1 diff ^1^			0.33	0.10	−0.03	−0.03
% diff ^2^			0.54	0.16	−1.30	−1.30
*p* trend			0.413	<0.001	0.039	<0.001
**Women (*n* = 2051)**	**Plasma α-Tocopherol, Adjusted for Cholesterol (μmol/mmol)**		**FFM**		**FFM_BMI_**	
Quintile	Mean ± SD	Median	Unadjusted	Adjusted	Unadjusted	Adjusted
1 (*n* = 411)	3.22 ± 0.39	3.32	40.41 ± 0.24	40.48 ± 0.23	1.58 ± 0.01	1.58 ± 0.01
2 (*n* = 410)	3.86 ± 0.13	3.87	40.19 ± 0.23	40.22 ± 0.23	1.56 ± 0.01	1.56 ± 0.01
3 (*n* = 410)	4.31 ± 0.13	4.30	40.34 ± 0.24	40.33 ± 0.23	1.54 ± 0.01 *	1.54 ± 0.01 *
4 (*n* = 410)	4.79 ± 0.16	4.79	40.75 ± 0.22	40.70 ± 0.23	1.57 ± 0.01	1.57 ± 0.01
5 (*n* = 410)	5.96 ± 1.03	5.61	40.58 ± 0.23	40.53 ± 0.23	1.55 ± 0.01	1.55 ± 0.01
Q5-Q1 diff ^1^			0.17	0.05	−0.03	−0.03
% diff ^2^			0.42	0.12	−1.90	−1.90
*p* trend			0.251	0.006	0.239	<0.001
**Men (*n* = 2232)**	**Plasma γ-Tocopherol, Adjusted for Cholesterol (μmol/mmol)**		**FFM**		**FFM_BMI_**	
Quintile	Mean ± SD	Median	Unadjusted	Adjusted	Unadjusted	Adjusted
1 (*n* = 447)	0.15 ± 0.04	0.16	60.78 ± 0.28	60.94 ± 0.24	2.32 ± 0.01	2.32 ± 0.01
2 (*n* = 446)	0.22 ± 0.01	0.22	60.96 ± 0.26	61.29 ± 0.24	2.33 ± 0.01	2.32 ± 0.01
3 (*n* = 447)	0.28 ± 0.02	0.28	61.27 ± 0.28	61.24 ± 0.24	2.30 ± 0.01	2.30 ± 0.01
4 (*n* = 446)	0.34 ± 0.02	0.34	61.58 ± 0.27 *	61.53 ± 0.24	2.30 ± 0.01	2.31 ± 0.01
5 (*n* = 446)	0.51 ± 0.14	0.47	61.94 ± 0.27 **	61.52 ± 0.24	2.27 ± 0.01 **	2.27 ± 0.01 **
Q5-Q1 diff ^1^			1.16	0.58	−0.05	−0.05
% diff ^2^			1.91	0.95	−2.16	−2.16
*p* trend			0.0007	*p* < 0.0001	0.0009	*p* < 0.0001
**Women (*n* = 2051)**	**Plasma γ-Tocopherol, Adjusted for Cholesterol (μmol/mmol)**		**FFM**		**FFM_BMI_**	
Quintile	Mean ± SD	Median	Unadjusted	Adjusted	Unadjusted	Adjusted
1 (*n* = 411)	0.14 ± 0.04	0.14	39.94 ± 0.22	39.93 ± 0.23	1.58 ± 0.01	1.58 ± 0.01
2 (*n* = 410)	0.21 ± 0.02	0.21	40.40 ± 0.24	40.39 ± 0.23	1.58 ± 0.01	1.58 ± 0.01
3 (*n* = 410)	0.26 ± 0.02	0.26	40.50 ± 0.23	40.52 ± 0.23	1.57 ± 0.01	1.57 ± 0.01
4 (*n* = 410)	0.33 ± 0.02	0.33	40.81 ± 0.23 **	40.82 ± 0.23 **	1.54 ± 0.01 *	1.54 ± 0.01 *
5 (*n* = 410)	0.48 ± 0.12	0.44	40.62 ± 0.23 *	40.61 ± 0.23 *	1.54 ± 0.01 **	1.54 ± 0.01 **
Q5-Q1 diff ^1^			0.68	0.68	−0.04	−0.04
% diff^2^			1.70	1.70	−2.53	−2.53
*p* trend			0.0254	0.0017	0.0013	*p* < 0.0001
**Men (*n* = 2232)**	**Plasma α:γ-Tocopherol Ratio, Adjusted for Cholesterol**		**FFM**		**FFM_BMI_**	
Quintile	Mean ± SD	Median	Unadjusted	Adjusted	Unadjusted	Adjusted
1 (*n* = 447)	8.73 ± 1.48	8.99	62.22 ± 0.27	61.93 ± 0.24	2.29 ± 0.01	2.29 ± 0.01
2 (*n* = 446)	12.13 ± 0.82	12.15	61.04 ± 0.26 **	60.97 ± 0.24 **	2.29 ± 0.01	2.29 ± 0.01
3 (*n* = 447)	15.04 ± 0.90	14.96	61.37 ± 0.28 *	61.39 ± 0.24	2.32 ± 0.01	2.32 ± 0.01
4 (*n* = 446)	19.01 ± 1.46	18.87	61.11 ± 0.27 **	61.24 ± 0.24 *	2.31 ± 0.01	2.31 ± 0.01
5 (*n* = 446)	34.97 ± 30.04	26.65	60.78 ± 0.28 ***	61.00 ± 0.24 **	2.31 ± 0.01	2.31 ± 0.01
Q5-Q1 diff ^1^			−1.44	−0.93	0.02	0.02
% diff ^2^			−2.31	−1.50	0.87	0.87
*p* trend			0.002	*p* < 0.001	0.109	*p* < 0.001
**Women (*n* = 2051)**	**Plasma α:γ-Tocopherol Ratio, Adjusted for Cholesterol**		**FFM**		**FFM_BMI_**	
Quintile	Mean ± SD	Median	Unadjusted	Adjusted	Unadjusted	Adjusted
1 (*n* = 411)	9.11 ± 1.60	9.44	40.51 ± 0.23	40.53 ± 0.23	1.54 ± 0.01	1.54 ± 0.01
2 (*n* = 410)	12.74 ± 0.90	12.78	40.87 ± 0.23	40.88 ± 0.23	1.55 ± 0.01	1.55 ± 0.01
3 (*n* = 410)	15.92 ± 1.04	15.83	40.37 ± 0.23	40.39 ± 0.23	1.55 ± 0.01	1.55 ± 0.01
4 (*n* = 410)	20.47 ± 1.74	20.31	40.30 ± 0.23	40.27 ± 0.23	1.59 ± 0.01 **	1.59 ± 0.01 **
5 (*n* = 410)	44.84 ± 50.09	31.67	40.22 ± 0.22	40.20 ± 0.23	1.58 ± 0.01 *	1.58 ± 0.01 *
Q5-Q1 diff ^1^			−0.29	−0.33	0.04	0.04
% diff ^2^			−0.72	−0.81	2.60	2.60
*p* trend			0.133	0.003	0.003	*p* < 0.001

Values are presented as means ± SEM. The *p*-trend was calculated using ANCOVA. ^1^ Q5-Q1 calculates the absolute difference between the means of quintile (Q) 5 and Q1. ^2^ % difference calculates the percentage difference between the means of Q5 and Q1. * *p* < 0.05; ** *p* < 0.01; *** *p* < 0.001 versus quintile 1. FFM = fat-free mass. For the FFM model, ratios were adjusted for age, smoking status, physical activity, corticosteroid and statin use, social class, fat mass, and menopausal status and HRT use in women. For the FFM_BMI_ model, ratios were adjusted for age, smoking status, physical activity, corticosteroid and statin use, social class, and menopausal status and HRT use in women.

**Table 5 antioxidants-10-00159-t005:** Associations between quintiles of dietary vitamin E and calcaneal BUA in men and women aged 42–82 years.

Men (*n* = 6443)	Dietary α-Tocopherol Equivalents Intake (mg/day)		BUA	
Quintile	Mean ± SD	Median	Unadjusted	Adjusted
1 (*n* = 1289)	6.45 ± 1.19	6.71	88.73 ± 0.50	89.03 ± 0.52
2 (*n* = 1289)	8.99 ± 0.53	9.01	90.06 ± 0.48	90.07 ± 0.49
3 (*n* = 1288)	10.93 ± 0.63	10.89	90.64 ± 0.48 **	90.64 ± 0.48 *
4 (*n* = 1289)	13.46 ± 0.90	13.38	90.51 ± 0.49 *	90.39 ± 0.49
5 (*n* = 1288)	19.71 ± 4.63	18.30	90.34 ± 0.49 *	90.15 ± 0.52
Q5-Q1 diff ^1^			1.61	1.12
% diff ^2^			1.81	1.26
*p* trend			0.044	<0.001
**Women (*n* = 8251)**				
Quintile	Mean ± SD	Median	Unadjusted	Adjusted
1 (*n* = 1651)	5.33 ± 0.98	5.54	70.85 ± 0.41	71.84 ± 0.39
2 (*n* = 1650)	7.38 ± 0.44	7.38	71.66 ± 0.40	72.16 ± 0.35
3 (*n* = 1650)	8.91 ± 0.46	8.93	71.13 ± 0.40	71.53 ± 0.35
4 (*n* = 1650)	10.72 ± 0.64	10.68	73.23 ± 0.41 ***	72.45 ± 0.35
5 (*n* = 1650)	15.13 ± 3.56	14.12	73.57 ± 0.41 ***	72.47 ± 0.38
Q5–Q1 diff ^1^			2.72	0.63
% diff ^2^			3.84	0.88
*p* trend			<0.001	<0.001

Values are presented as means ± SEM. The *p*-trend was calculated using ANCOVA. ^1^ Q5–Q1 calculates the absolute difference between the means of quintile (Q) 5 and Q1. ^2^ % difference calculates the percentage difference between the means of Q5 and Q1. * *p* < 0.05; ** *p* < 0.01; *** *p* < 0.001 versus quintile 1, according to ANCOVA. BUA = broadband ultrasound attenuation. BUA—adjusted model includes age, BMI, total energy, dietary calcium intake, dietary vitamin D intake, smoking status, physical activity, corticosteroid use, menopausal status, HRT use, calcium and vitamin D supplement use and family history of osteoporosis.

**Table 6 antioxidants-10-00159-t006:** Multivariate adjusted calcaneal BUA for EPIC-Norfolk participants, stratified by sex and quintiles of plasma α-tocopherol, plasma γ-tocopherol and α-tocopherol:γ-tocopherol ratio, adjusted for by blood cholesterol measurement.

Men (*n* = 2308)	Plasma α-Tocopherol, Adjusted for Cholesterol (μmol/mmol)		BUA	
Quintile	Mean ± SD	Median	Unadjusted	Adjusted
1 (*n* = 462)	3.17 ± 0.36	3.26	88.97 ± 0.82	89.11 ± 0.82
2 (*n* = 462)	3.80 ± 0.13	3.80	89.01 ± 0.81	88.87 ± 0.82
3 (*n* = 461)	4.24 ± 0.13	4.24	89.22 ± 0.83	89.01 ± 0.82
4 (*n* = 462)	4.72 ± 0.15	4.70	89.84 ± 0.83	89.84 ± 0.82
5 (*n* = 461)	5.90 ± 1.20	5.50	90.19 ± 0.84	90.38 ± 0.83
Q5–Q1 diff ^1^			1.22	1.27
% diff ^2^			1.37	1.42
*p* trend			0.209	<0.001
**Women (*n* = 2149)**	**Plasma α-Tocopherol, Adjusted for Cholesterol (μmol/mmol)**		**BUA**	
Quintile	Mean ± SD	Median	Unadjusted	Adjusted
1 (*n* = 430)	3.22 ± 0.41	3.33	68.74 ± 0.81	69.29 ± 0.66
2 (*n* = 430)	3.86 ± 0.13	3.87	69.22 ± 0.79	69.47 ± 0.66
3 (*n* = 430)	4.31 ± 0.13	4.30	69.38 ± 0.72	69.25 ± 0.66
4 (*n* = 430)	4.80 ± 0.16	4.80	70.86 ± 0.78	70.72 ± 0.66
5 (*n* = 429)	5.96 ± 1.02	5.63	70.95 ± 0.78 *	70.44 ± 0.67
Q5–Q1 diff ^1^			2.21	1.15
% diff ^2^			3.22	1.66
*p* trend			0.015	<0.001
**Men (*n* = 2308)**	**Plasma γ-Tocopherol, Adjusted for Cholesterol (μmol/mmol)**		**BUA**	
Quintile	Mean ± SD	Median	Unadjusted	Adjusted
1 (*n* = 462)	0.15 ± 0.04	0.16	88.93 ± 0.79	88.68 ± 0.83
2 (*n* = 462)	0.22 ± 0.02	0.23	89.16 ± 0.86	89.09 ± 0.82
3 (*n* = 461)	0.28 ± 0.02	0.28	89.10 ± 0.84	88.94 ± 0.82
4 (*n* = 462)	0.35 ± 0.02	0.34	89.50 ± 0.82	89.80 ± 0.82
5 (*n* = 461)	0.51 ± 0.14	0.47	90.54 ± 0.82	90.69 ± 0.82
Q5–Q1 diff ^1^			1.61	2.01
% diff ^2^			1.81	2.27
*p* trend			0.141	*p* < 0.001
**Women (*n* = 2149)**	**Plasma γ-Tocopherol, Adjusted for Cholesterol (μmol/mmol)**		**BUA**	
Quintile	Mean ± SD	Median	Unadjusted	Adjusted
1 (*n* = 430)	0.14 ± 0.04	0.14	69.45 ± 0.79	68.46 ± 0.67
2 (*n* = 430)	0.21 ± 0.02	0.21	69.22 ± 0.78	68.86 ± 0.66
3 (*n* = 430)	0.27 ± 0.02	0.26	68.93 ± 0.78	69.03 ± 0.66
4 (*n* = 430)	0.33 ± 0.02	0.33	70.34 ± 0.76	70.69 ± 0.66
5 (*n* = 429)	0.49 ± 0.13	0.44	71.21 ± 0.78	72.12 ± 0.66
Q5–Q1 diff ^1^			1.76	3.66
% diff ^2^			2.53	5.35
*p* trend			0.046	*p* < 0.001
**Men (*n* = 2308)**	**Plasma α:γ-Tocopherol Ratio, Adjusted for Cholesterol**		**BUA**	
Quintile	Mean ± SD	Median	Unadjusted	Adjusted
1 (*n* = 462)	8.73 ± 1.46	9.00	89.44 ± 0.82	89.67 ± 0.83
2 (*n* = 462)	12.10 ± 0.82	10.70	89.56 ± 0.81	89.67 ± 0.82
3 (*n* = 461)	14.97 ± 0.88	13.51	89.78 ± 0.85	89.76 ± 0.82
4 (*n* = 462)	18.91 ± 1.45	16.59	89.51 ± 0.82	89.31 ± 0.82
5 (*n* = 461)	35.15 ± 30.64	21.69	88.95 ± 0.82	88.79 ± 0.83
Q5–Q1 diff ^1^			−0.49	−0.88
% diff ^2^			−0.55	−0.98
*p* trend			0.602	*p* < 0.001
**Women (*n* = 2149)**	**Plasma α:γ-Tocopherol Ratio, Adjusted for Cholesterol**		**BUA**	
Quintile	Mean ± SD	Median	Unadjusted	Adjusted
1 (*n* = 430)	9.06 ± 1.62	9.42	70.31 ± 0.79	71.02 ± 0.66
2 (*n* = 430)	12.67 ± 0.90	12.75	70.03 ± 0.74	70.97 ± 0.66
3 (*n* = 430)	15.81 ± 1.02	15.75	69.32 ± 0.78	69.31 ± 0.66
4 (*n* = 430)	20.35 ± 1.76	20.13	69.68 ± 0.80	69.10 ± 0.66
5 (*n* = 429)	44.22 ± 57.93	31.23	69.81 ± 0.79	68.74 ± 0.68
Q5–Q1 diff ^1^			−0.50	−2.28
% diff ^2^			−0.71	−3.21
*p* trend			0.731	*p* < 0.001

Values are presented as means ± SEM. The *p*-trend was calculated using ANCOVA. BUA = broadband ultrasound attenuation. ^1^ Q5–Q1 calculates the absolute difference between the means of quintile (Q) 5 and Q1. ^2^ % difference calculates the percentage difference between the means of Q5 and Q1. * *p* < 0.05 versus quintile 1. Ratios were adjusted for age, BMI, smoking status, physical activity, family history of osteoporosis, corticosteroid use, dietary calcium intake, dietary vitamin D intake, calcium and vitamin D supplement use, and menopausal status and HRT use in women.

**Table 7 antioxidants-10-00159-t007:** Risk of total, hip, spine and wrist fractures in EPIC-Norfolk participants, stratified by sex and quintiles of dietary vitamin E intake (*n* = 25,223).

	Men (*n* = 11,427)					Total Fractures				Hip Fracture					Spine Fracture					Wrist Fracture			
		N	Mean	SD	Median	Incidence	HR	95% CI		Incidence		HR	95% CI		Incidence		HR	95% CI		Incidence	HR	95% CI	
Dietary α-tocopherol equivalents intake (mg/day)	Q1	2286	5.95	1.25	6.23	188/2286	1.00	Ref.		80/2286		1.00	Ref.		49/2286		1.00	Ref.		33/2286	1.00	Ref.	
	Q2	2285	8.65	0.59	8.67	161/2285	0.79 *	0.64	0.98	74/2285		0.89	0.64	1.23	47/2285		0.94	0.62	1.41	20/2285	0.51 *	0.29	0.90
	Q3	2286	10.65	0.61	10.62	176/2286	0.94	0.76	1.17	62/2286		0.86	0.61	1.22	49/2286		1.10	0.73	1.67	32/2286	0.80	0.48	1.34
	Q4	2285	13.22	0.92	13.11	158/2285	0.86	0.68	1.08	72/2285		1.08	0.76	1.53	33/2285		0.77	0.48	1.24	38/2285	0.88	0.53	1.47
	Q5	2285	19.71	4.92	18.22	194/2285	1.06	0.84	1.34	68/2285		1.05	0.73	1.52	45/2285		1.09	0.68	1.75	32/2285	0.68	0.38	1.19
						877/11,427				356/11,427					223/11,427					155/11,427			
*p* trend							*p* < 0.001					*p* < 0.001					0.21				0.11		
	**Women (*n* = 13,796)**					**Total Fractures**				**Hip Fracture**					**Spine Fracture**					**Wrist Fracture**			
		N	Mean	SD	Median	Incidence	HR	95% CI		Incidence		HR	95% CI		Incidence		HR	95% CI		Incidence	HR	95% CI	
Dietary α-tocopherol equivalents intake (mg/day)	Q1	2760	5.04	1.03	5.25	479/2760	1.00	Ref.		228/2760		1.00	Ref.		89/2760		1.00	Ref.		108/2760	1.00	Ref.	
	Q2	2759	7.14	0.45	7.15	438/2759	0.93	0.81	1.06	177/2759		0.81 *	0.66	0.99	80/2759		0.94	0.68	1.28	104/2759	1.05	0.80	1.39
	Q3	2759	8.70	0.46	8.68	427/2759	0.93	0.81	1.07	212/2759		1.04	0.85	1.27	63/2759		0.77	0.55	1.09	102/2759	1.08	0.81	1.44
	Q4	2759	10.55	0.66	10.50	377/2759	0.88	0.76	1.02	175/2759		0.93	0.75	1.16	67/2759		0.90	0.63	1.28	104/2759	1.20	0.89	1.63
	Q5	2759	15.01	3.48	14.01	371/2759	0.90	0.77	1.06	179/2759		1.02	0.81	1.29	58/2759		0.82	0.55	1.21	86/2759	1.07	0.77	1.50
						2092/13,796				971/13,796					357/13,796					504/13,796			
*p* trend							*p* < 0.001					*p* < 0.001					0.09				*p* < 0.01		

* *p* < 0.05 versus quintile 1. Ratios were adjusted for age, BMI, smoking status, physical activity, family history of osteoporosis, calcium intake, vitamin D intake, energy intake, calcium- and vitamin D-containing supplement use, corticosteroid use, and menopausal status and HRT use in women.

**Table 8 antioxidants-10-00159-t008:** Risk of total, hip, spine and wrist fractures in EPIC-Norfolk participants, stratified by sex and quintiles of plasma α-tocopherol, plasma γ-tocopherol and the ratio of α:γ, adjusted for cholesterol.

Men (*n* = 3727)				Total Fractures				Hip Fractures				Spine Fractures				Wrist Fractures			
	Mean	SD	Median	Incidence	HR	95% CI		Incidence	HR	95% CI		Incidence	HR	95% CI		Incidence	HR	95% CI	
Plasma α-tocopherol, adjusted for cholesterol (μmol/mmol)	3.13	0.39	3.23	67/746	1.00	Ref.		31/746	1.00	Ref.		20/746	1.00	Ref.		4/746	1.00	Ref.	
	3.79	0.13	3.79	52/745	0.79	0.54	1.13	20/745	0.62	0.35	1.11	16/745	0.84	0.43	1.62	10/745	2.41	0.75	7.72
	4.22	0.13	4.22	61/746	0.90	0.63	1.27	25/746	0.82	0.48	1.39	18/746	0.87	0.46	1.65	10/746	2.29	0.71	7.35
	4.70	0.16	4.69	65/745	0.88	0.63	1.24	27/745	0.78	0.47	1.32	17/745	0.78	0.41	1.50	8/745	1.76	0.52	5.87
	5.88	1.12	5.51	63/745	0.86	0.61	1.23	26/745	0.79	0.46	1.35	18/745	0.81	0.42	1.56	8/745	1.79	0.53	6.05
*p* trend					0.028				0.007				0.460				0.787		
Plasma γ-tocopherol, adjusted for cholesterol (μmol/mmol)	0.15	0.04	0.16	60/746	1.00	Ref.		23/746	1.00	Ref.		20/746	1.00	Ref.		7/746	1.00	Ref.	
	0.23	0.02	0.23	52/745	0.85	0.58	1.24	24/745	1.03	0.58	1.85	13/745	0.63	0.31	1.28	9/745	1.25	0.46	3.39
	0.28	0.02	0.28	61/746	0.97	0.68	1.40	27/746	1.09	0.62	1.94	21/746	1.00	0.54	1.85	6/746	0.83	0.28	2.53
	0.35	0.02	0.35	75/745	1.21	0.85	1.71	30/745	1.22	0.69	2.14	21/745	0.98	0.53	1.83	9/745	1.28	0.47	3.49
	0.52	0.13	0.48	60/745	0.94	0.65	1.37	25/745	1.04	0.57	1.86	14/745	0.60	0.29	1.22	9/745	1.31	0.47	3.63
*p* trend					0.029				0.008				0.425				0.776		
Plasma α-tocopherol:γ-tocopherol ratio, adjusted for cholesterol	8.56	1.43	8.87	60/746	1.00	Ref.		28/746	1.00	Ref.		13/746	1.00	Ref.		7/746	1.00	Ref.	
	11.90	0.77	11.93	64/745	1.00	0.70	1.42	31/745	0.94	0.56	1.57	19/745	1.53	0.74	3.17	5/745	0.67	0.21	2.12
	14.71	0.87	14.66	76/746	1.18	0.84	1.66	28/746	0.85	0.50	1.45	25/746	2.00*	1.00	4.00	11/746	1.42	0.55	3.68
	18.53	1.39	18.40	50/745	0.86	0.59	1.26	15/745	0.55	0.29	1.04	19/745	1.65	0.79	3.43	8/745	1.13	0.40	3.16
	34.08	27.74	25.88	58/745	0.96	0.66	1.40	27/745	0.96	0.55	1.68	13/745	1.09	0.49	2.44	9/745	1.24	0.44	3.45
*p* trend					0.027				0.007				0.490				0.765		
**Women (*n* = 3564)**				**Total Fractures**				**Hip Fractures**				**Spine Fracture**				**Wrist Fracture**			
	Mean	SD	Median	Incidence	HR	95% CI		Incidence	HR	95% CI		Incidence	HR	95% CI		Incidence	HR	95% CI	
Plasma α-tocopherol, adjusted for cholesterol (μmol/mmol)	3.20	0.42	3.32	138/713	1.00	Ref.		66/713	1.00	Ref.		24/713	1.00	Ref.		37/713	1.00	Ref.	
	3.86	0.13	3.86	124/713	0.87	0.68	1.11	66/713	0.99	0.70	1.39	25/713	1.05	0.60	1.84	26/713	0.67	0.40	1.12
	4.30	0.12	4.29	125/713	0.98	0.77	1.25	62/713	1.08	0.76	1.52	23/713	1.04	0.58	1.84	34/713	0.98	0.61	1.56
	4.78	0.17	4.77	132/713	1.03	0.81	1.31	56/713	0.93	0.65	1.33	27/713	1.20	0.69	2.10	36/713	1.04	0.65	1.65
	5.97	1.04	5.61	124/712	0.94	0.73	1.20	63/712	1.04	0.73	1.49	25/712	1.03	0.57	1.85	19/712	0.56 *	0.32	0.98
*p* trend					0.010				0.013				0.665				0.075		
Plasma γ-tocopherol, adjusted for cholesterol (μmol/mmol)	0.14	0.04	0.15	125/713	1.00	Ref.		62/713	1.00	Ref.		26/713	1.00	Ref.		32/713	1.00	Ref.	
	0.21	0.02	0.21	132/713	1.11	0.87	1.42	60/713	1.01	0.71	1.45	23/713	0.91	0.52	1.62	31/713	1.04	0.63	1.71
	0.27	0.02	0.27	125/713	1.09	0.84	1.40	57/713	0.97	0.67	1.41	29/713	1.13	0.65	1.96	33/713	1.17	0.71	1.92
	0.33	0.02	0.33	143/713	1.29 *	1.01	1.65	76/713	1.36	0.96	1.93	28/713	1.20	0.69	2.09	30/713	1.04	0.62	1.75
	0.50	0.13	0.46	118/712	1.07	0.82	1.39	58/712	1.05	0.72	1.52	18/712	0.75	0.40	1.40	26/712	1.00	0.58	1.70
*p* trend					0.009				0.011				0.486				0.108		
Plasma α-tocopherol:γ-tocopherol ratio, adjusted for cholesterol	8.91	1.66	9.30	131/713	1.00	Ref.		69/713	1.00	Ref.		23/713	1.00	Ref.		29/713	1.00	Ref.	
	12.53	0.89	12.58	130/713	0.96	0.76	1.23	70/713	1.01	0.72	1.41	23/713	1.02	0.57	1.83	23/713	0.73	0.42	1.28
	15.64	0.99	15.56	123/713	0.90	0.70	1.15	43/713	0.61 *	0.41	0.89	29/713	1.30	0.75	2.26	42/713	1.40	0.87	2.26
	20.06	1.70	19.87	138/713	0.98	0.77	1.25	71/713	0.97	0.69	1.36	25/713	1.02	0.57	1.82	31/713	0.93	0.56	1.56
	42.51	50.07	30.12	121/712	0.84	0.65	1.10	60/712	0.83	0.58	1.19	24/712	0.97	0.53	1.79	27/712	0.84	0.48	1.44
*p* trend					0.007				0.011				0.507				0.099		

* *p* < 0.05 versus quintile 1. Ratios were adjusted for age, BMI, smoking status, physical activity, family history of osteoporosis, calcium intake, vitamin D intake, calcium- and vitamin D-containing supplement use, corticosteroid use and menopausal status and HRT use in women.

**Table 9 antioxidants-10-00159-t009:** Summary of the associations of dietary intake and plasma concentrations of vitamin E with SMM, BUA and fracture risk.

Measure of Interest		Dietary α-Tocopherol Equivalents Intake (mg/day)	% Difference ^1^	Serum Cholesterol-Adjusted α-Tocopherol (μmol/mmol)	% Difference ^1^	Serum Cholesterol-Adjusted γ-Tocopherol (μmol/mmol)	% Difference ^1^	Serum Cholesterol-Adjusted α-tocopherol:γ-Tocopherol Ratio	% Difference ^1^
FFM	Men	positive ***	1.03	positive ***	0.16	positive ***	0.95	negative ***	−1.50
	Women	positive ***	1.04	positive **	0.12	positive **	1.70	negative **	−0.81
FFM_BMI_	Men	positive ***	1.74	negative ***	−1.30	negative ***	−2.16	positive ***	0.87
	Women	positive ***	1.91	negative ***	−1.90	negative ***	−2.53	positive ***	2.60
BUA	Men	positive ***	1.26	positive ***	1.42	positive ***	2.27	negative ***	−0.98
	Women	positive ***	0.88	positive ***	1.66	positive ***	5.35	negative ***	−3.21
Total fracture risk ^2^	Men	positive ***		positive *		positive *		positive *	
	Women	positive ***		positive *		negative **		positive **	
Hip fracture risk ^2^	Men	positive ***		positive **		negative **		positive **	
	Women	positive ***		positive *		negative *		positive *	
Spine fracture risk ^2^	Men	N.S.		N.S.		N.S.		N.S.	
	Women	N.S.		N.S.		N.S.		N.S.	
Wrist fracture risk ^2^	Men	N.S.		N.S.		N.S.		N.S.	
	Women	negative **		N.S.		N.S.		N.S.	

FFM = fat-free mass; BUA = broadband ultrasound attenuation. ^1^ (Q5 mean − Q1 mean/Q1 mean) × 100. ^2^ Positive associations with fracture risk indicate lower risk with higher tocopherol, while negative associations indicate higher risk. * *p* < 0.05; ** *p* < 0.01; *** *p* < 0.001.

## Data Availability

The authors will make the dataset available under a Data Transfer Agreement to any bona fide researcher who wishes to obtain the dataset in order to undertake a replication analysis. Although the dataset is anonymized, the breadth of the data included and the multiplicity of variables that are included in this analysis file as primary variables or confounding factors, means that provision of the dataset to other researchers without a Data Transfer Agreement would constitute a risk. Requests for data sharing/access should be submitted to the EPIC Management Committee (epic-norfolk@mrc-epid.cam.ac.uk).

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
