# Peer review of "Positive Associations of Dietary Intake and Plasma Concentrations of Vitamin E with Skeletal Muscle Mass, Heel Bone Ultrasound Attenuation and Fracture Risk in the EPIC-Norfolk Cohort"

_antioxidants, 2021, doi:10.3390/antiox10020159_

Round 1

Reviewer 1 Report

Dear Authors, 

Congratulations for your large study. It is very interesting and very well done.I want to insert a paragraph regarding the limitation of your study and also to explain if it is suitable to recommend vitamin E supplements for younger people just to avoid the fracture risk and sarcopenia. In line with this topic, I recommend inserting also a possible recommendation for vitamin E supplementation for the people with the sarcopenia risk.

For improving your manuscript I recommend :

-line 125: explain abbreviation 'DEXA' and use the abbreviation in line 590  

-line 473 : use "study"/manuscript instead of paper
-line 573 : please insert a paragraph regarding the limitation of your study.

-line 592 : explain if it is suitable to recommend vitamin E supplements for younger people just to avoid fracture risk and sarcopenia. 

-line 592: In line with this topic, I recommend inserting also a possible recommendation for vitamin E supplementation for the people with the sarcopenia risk.  

Reviewer 2 Report

The manuscript is well-written and data are of certain importance.

A few minor points:

'Ps' are uppercase in the text and lowercase in the table. Choose one or the other and be consistent.

Line 16: skeletal muscle mass

Line 64: You need 3-4 sentences to make a paragraph.

Line 102: at a second

Line 123: BMD assessment

Line 125: add abbreviation 'DEXA' and use the abbreviation in line 590

Line 473: this study not this paper

Line 497: focused

Please spell out the following 7dDDs 24hDRs FFQs

Reviewer 3 Report

The study by Mulligan et al. provides an interesting insight into the possible role of vitamin E in sarcopenia and osteoporosis. A major strength of the study is the number of participants who were included into the study, which increases the validity and clinical relevance of the data. However, there are some issues that should be addressed:

  • Vitamin D has a major role in bone (as well as skeletal muscle health), but its concentration was not determined. This is really a pity because it would be of major importance for interpretation of the data. The role and potential impact of vitamin D should be discussed in more detail in the Discussion.
  • Follow-up was approximately 18 years, a long period during which physical activity and dietary habits, including vitamin E intake, and medical therapies (e.g. corticosteroids, HRT, vitamin D supplementation etc.) likely changed. This aspect, which is highly relevant for the fracture data, should be highlighted in the Discussion.
  • Vitamin D and vitamin E are both fat-soluble vitamins with similar modes of absorption. In addition, they may be taken together in multivitamin supplements. It would be important to determine whether vitamin D and calcium intake correlated with vitamin E intake and especially vitamin E plasma concentrations. Perhaps participants with higher vitamin E concentrations also ingested more vitamin D?
  • Tables 1 and 2: were there any statistically significant differences (for any of the parameters) between the three study groups?
  • Table 4: were any adjustments for confounding variables made here?
  • Analysis of bone density status in Table 5b apparently did not take into account calcium or vitamin D intake. In contrast, statistical model was adjusted for these variables in Table 3b. If the model in Table 5b did not take into account calcium and vitamin D, statistical analysis should be repeated by including these confounding variables in the model.
  • Analysis of fracture risk in Table 6 apparently did not take into account calcium and vitamin D intake or vitamin D plasma concentrations. Fracture risk should take into consideration these parameters.
  • Statistical analysis was adjusted for corticosteroid use. Does this mean corticosteroid use at baseline or at the time of fracture? Since the follow-up period was long some patients were likely started on corticosteroids after inclusion into the study.
  • In the current version of the manuscript data for dietary intake of vitamin E and vitamin E plasma concentrations are presented separately. I suggest the authors simplify their presentation and present correlations of vitamin E intake and plasma concentrations together for each of the clinical outcomes. It would be easier to follow the text if these results are presented together rather than in six separate sections (twice for SSM, twice for bone density, twice for fractures).
  • “Muscle is now recognised to have paracrine and endocrine effects, and that myokines secreted by muscle, including insulin-48 like growth factor 1 and fibroblast growth factor 2, can affect bone repair and metabolism[10,11].” It would probably be better to say that factors (myokines) secreted from skeletal muscle have paracrine and endocrine effects rather than that the muscle has such effects.
  • It would be nice to include a proposed mechanism by which vitamin E may protect against sarcopenia and osteoporosis (in the Discussion).

Round 2

Reviewer 3 Report

The authors have addressed all my questions and concerns. Additional statistical analyses and amendments to the text have improved the manuscript and clarified open questions. I do not have any further comments.